# L-Proline Activates Mammalian Target of Rapamycin Complex 1 and Modulates Redox Environment in Porcine Trophectoderm Cells

**DOI:** 10.3390/biom11050742

**Published:** 2021-05-17

**Authors:** Ning Liu, Ying Yang, Xuemeng Si, Hai Jia, Yunchang Zhang, Da Jiang, Zhaolai Dai, Zhenlong Wu

**Affiliations:** 1State Key Laboratory of Animal Nutrition, Department of Animal Nutrition and Feed Science, China Agricultural University, Beijing 100193, China; nuli982390@163.com (N.L.); cauvet2020@outlook.com (Y.Y.); sxmswun@126.com (X.S.); jiahai@cau.edu.cn (H.J.); B20173040318@cau.edu.cn (Y.Z.); B20193040341@cau.edu.cn (D.J.); daizhaolai@cau.edu.cn (Z.D.); 2Beijing Advanced Innovation Center for Food Nutrition and Human Health, China Agricultural University, Beijing 100193, China

**Keywords:** L-proline, mammalian target of rapamycin complex 1, porcine trophectoderm cell line 2, proline dehydrogenase, transporters, redox homeostasis

## Abstract

L-proline (proline) is a key regulator of embryogenesis, placental development, and fetal growth. However, the underlying mechanisms that support the beneficial effects of proline are largely unknown. This study used porcine trophectoderm cell line 2 (pTr2) to investigate the underlying mechanisms of proline in cell proliferation and redox homeostasis. Cells were cultured in the presence of 0, 0.25, 0.50, or 1.0 mmol/L proline for an indicated time. The results showed that 0.5 and 1.0 mmol/L proline enhanced cell viability. These effects of proline (0.5 mmol/L) were accompanied by the enhanced protein abundance of p-mTORC1, p-p70S6K, p-S6, and p-4E-BP1. Additionally, proline dose-dependently enhanced the mRNA expression of proline transporters [solute carrier family (*SLC*) *6A20*, *SLC36A1*, *SLC36A2*, *SLC38A1*, and *SLC38A2*], elevated proline concentration, and protein abundance of proline dehydrogenase (PRODH). Furthermore, proline addition (0.25 or 0.5 mmol/L) resulted in lower abundance of p-AMPK*α* when compared with a control. Of note, proline resulted in lower reactive oxygen species (ROS) level, upregulated mRNA expression of the catalytic subunit of glutamate–cysteine ligase (*GCLC*) and glutathione synthetase (*GSS*), as well as enhanced total (T)-GSH and GSH concentration when compared with a control. These data indicated that proline activates themTORC1 signaling and modulates the intracellular redox environment via enhancing proline transport.

## 1. Introduction

Embryonic loss and fetal death during the gestation period remain a serious reproductive problem in humans and animals [1,2,3]. In pigs, implantation of the conceptus starts on approximately day 13 and embryos undergo huge morphological changes to ensure conceptus maternal communication is built between conceptus trophectoderm and uterine luminal epithelium, which is pivotal for ensuring viability and successful implantation of the conceptus [4,5]. Indeed, proliferation, migration, and differentiation of trophectoderm cells play a key role in preparing for implantation and placentation of the conceptus [6,7]. Furthermore, exchange of nutrients, gases, hormones, growth factors, and other molecules across the placenta is essential for fetal growth and development [8]. Consequently, impairment of placental growth has been reported to contribute to intrauterine growth retardation (IUGR), a major problem in both human pregnancy and animal production [9]. Of particular interest, the trophectoderm, leading predominantly to extraembryonic tissues, including the placenta, were stimulated by arginine and putrescine through the mammalian target of rapamycin complex 1 (mTORC1) signaling pathway [9,10]. Porcine trophectoderm cell line 2 (pTr2) is a cell line model widely used in reproductive biology research [9,11,12].

Evidence supporting the early development of embryo occurs in a relatively low oxygen environment has been documented in studies of implantation [13]. The embryo is sensitive to oxidative injury because of its low antioxidant capacity [14,15,16,17]. Furthermore, elevated oxygen consumption and reactive oxidative species (ROS) level have been observed in the trophectoderm of the mouse and rabbit [18,19]. As placentation progresses, placental antioxidant enzyme concentrations are low during early pregnancy, but that they increase with gestational age, concurrent with increased oxygen transfer and the resulting cellular generation of ROS [17,20]. Previous evidence have linked excessive of ROS generation to developmental arrest and damage in embryos, as well as a reduction in placenta perfusion [13,14,21]. In addition, studies suggested that ROS regulate the proliferation and differentiation of porcine trophectoderm cells based on the number of cells of pTr2 that were changed by modulation of ROS [12,22]. Furthermore, several studies have revealed that regulation of intra- and extra-cellular reducing environments by reducing ROS with the use of antioxidants is effective for improving embryo development in vivo or in vitro [13,14,21,23].

L-Proline (proline), a multifunctional imino acid involved in cell metabolism and physiology, plays versatile roles in oxidative stress and osmotic protection, protein chaperoning, cell signaling, programmed cell death, and nutrient adaptation and survival [24,25,26,27]. Moreover, it serves as a nitrogenous substrate for endogenous synthesis of arginine [28] and polyamines [29], both of which play a positive role in porcine trophectoderm cell proliferation and migration, suggesting the importance of nutrients for normal development of the conceptus and implantation [9,10,11]. We have recently reported that maternal proline supplementation during gestation enhanced fetal survival, reproductive performance, placental growth, and amino acid transport from dam to fetus [30]. Of particular interest, results of recent studies indicated that proline was observed to reduce the intracellular ROS level of mammalian cells or oocytes [26,27,31]. Indeed, numerous laboratories proved that addition of proline led to enhanced total GSH and protection of intracellular reduced GSH through reaction with singlet oxygen (^1^O_2_), H_2_O_2_, and OH• (pH 7–8) to form stable free radical adducts of proline and hydroxyproline derivatives [32,33,34].

Based on the foregoing, we conducted a study with pTr2 cells to test the hypothesis that proline could activate the mTORC1 signaling pathway and increase intracellular GSH concentration.

## 2. Materials and Methods

### 2.1. Materials

Dulbecco’s modified Eagle’s medium (DMEM)/Ham’s F-12 (DMEM/F-12), fetal bovine serum (FBS), penicillin–streptomycin (PS), and customized DMEM were purchased from Gibco BRL company (Waltham, MA, USA). Antibodies against mammalian target of rapamycin complex 1 (mTORC1, catalog no. #2972), phosphorylated (p)-mTORC1 (catalog no. #2971), eukaryotic translation initiation factor 4E-binding protein 1 (4E-BP1, catalog no. #9452), p-4E-BP1 (catalog no. #13396), ribosomal protein S6 kinase (p70S6K, catalog no. #9202), p-p70S6K (catalog no. #9234), S6 (catalog no. #2217), p-S6 (catalog no. #2211), and AMP-activated protein kinase (AMPK, catalog no. #2532), and p-AMPK (catalog no. #2535) were products of Cell Signaling Technology (Beverly, MA, USA). Antibody against proline dehydrogenase (PRODH, catalog no. ab203875) was procured from Abcam (Cambridge, UK). Antibody against glyceraldehyde-3-phosphate dehydrogenase (GAPDH, catalog no. sc-59540) was purchased from Santa Cruz Biotechnology (Santa Cruz, CA, USA). Peroxidase-conjugated goat anti-rabbit and goat anti-mouse secondary antibodies were obtained from Huaxingbio Biotechnology Co. (Beijing, China). 2′,7′-dichlorofluorescein diacetate (DCFH-DA) was purchased from Beyotime Institute of Biotechnology (Jiangsu, China). Unless otherwise indicated, all other chemicals used in this study were obtained from Sigma-Aldrich (St. Louis, MO, USA).

### 2.2. Cell Culture

The pTr2 cells, a cell line (provided by Dr. Fuller W. Bazer, Texas A&M University) established from elongated porcine blastocysts, were cultured as previously described [9]. Briefly, the cells were cultured in 100 mm tissue culture dishes containing 8 mL DMEM/F-12 with 5% FBS, 1% PS, and 0.05% insulin in a standard humidified incubator at 37 °C and 5% CO_2_. Following the starvation period, cells were starved for 6 h without serum and insulin, and then incubated in the presence of 0, 0.25, 0.50, or 1.0 mmol/L proline for the indicated time periods.

### 2.3. Cell Viability Assay

The pTr2 cells (5000/well) were subcultured in a 96-well plate in DMEM/F-12 medium containing 5% FBS and 0.05% insulin and allowed to grow for 20 h, and then were starved for 6 h in serum- and insulin-free customized DMEM. After that, cells were incubated in the presence of various treatments for the indicated time. Cell viability was determined by adding Cell Counting Kit-8 (CCK-8, Zoman Biotech, Beijing, China) to each well.

### 2.4. Determination of ROS

Intracellular ROS levels were determined by a flow cytometer with the use of DCFH-DA, an ROS-sensitive probe, as previous described [35]. Briefly, cells were treated and incubated with DCFH-DA (10 μmol/L) at 37 °C for 30 min. ROS level was measured with a flow cytometer (Beckman Coulter, Brea, CA, USA). Data were assessed using CytExpert software (Beckman Coulter, Brea, CA, USA).

### 2.5. Analysis of GSH and Proline Concentration

Briefly, pTr2 cells in the 60 mm tissue culture dishes were collected, washed twice with PBS, and lysed with 0.1 mL of 1.5 mol/L HClO_4_. After incubation at 4 °C for 30 min, the solution was neutralized with 50 μL of 2 mol/L K_2_CO_3_. The neutralized supernatant fluids were analyzed by HPLC (Waters Inc., Milford, MA, USA) as described previously [30,36,37,38]. For GSH detection, washed cells with PBS were lysed in 0.1 mL of homogenization buffer (containing 6 mmol/L iodoacetic acid and 1.5 mol/L HClO_4_). After incubation, samples were neutralized with 50 μL of 2 mol/L K_2_CO_3_. GSH concentrations in the neutralized supernatant fluids were analyzed by using HPLC method as previously described [39]. These data were calculated by dividing the protein concentration that was measured by a bicinchoninic acid protein assay (BCA) assay.

### 2.6. Quantitative PCR Analysis

The qRT-PCR assay was performed as described previously [35,40]. Briefly, total RNA was isolated from cells by using the Trizol reagent (Cwbio, Nanjing, China) according to the producer’s information, and reverse-transcribed into cDNA by using a Fast Quant RT Kit (with gDNase). The qPCR was carried out with the SYBR Green mix and specific primers for mouse genes by using ABI 7500 real-time PCR system to determine the mRNA level of genes. The primer sequences used in this study can be found in Table 1. Fold changes were calculated using the 2^−ΔΔCt^ method, and GAPDH was used as a reference gene for normalization.

### 2.7. Western Blot Analyses

The preparation of cell lysates and the process of Western blot analysis were performed as previously described [41,42]. Briefly, harvested cells were lysed in ice-cold radioimmunoprecipitation (RIPA) lysis buffer containing 50 mmol/L Tris–HCl (pH 7.4), 150 mmol/L NaCl, 1% NP-40, 0.1% SDS, 1.0 mmol/L PMSF, 1.0 mmol/L Na_3_VO_4_, and a protease inhibitor tablet (Roche, Indianapolis, IN, USA). Supernatants were collected after centrifugation at 12,000 rpm for 10 min and protein concentration determined with a BCA kit (Huaxingbio, Beijing, China). Equal amounts of denatured proteins (25 μg) were separated using SDS-PAGE gels, transferred to polyvinylidene fluoride membranes (PVDF, Millipore, Billerica, MA, USA), and then were blocked in buffer (5% non-fat milk buffer for non-phosphorylated proteins and 5% bovine serum albumin buffer for phosphorylated proteins) at 25 °C for 1 h. The membranes were incubated with a primary antibody overnight at 4 °C, followed by incubation with an appropriate secondary antibody at 25 °C for 1 h. The images of protein bands were developed in an ImageQuant LAS 4000 mini system (GE Healthcare, Piscataway, NJ, USA) after incubation with enhanced chemiluminescence detection reagents (Huaxingbio, Beijing, China). Image Pro Plus 6.0 software (Media Cybernetics, CA, USA) was used to quantify the band density. GAPDH was used as the internal control.

### 2.8. Statistical Analysis

Values were presented as means ± SEMs and were statistically analyzed using GraphPad PRISM version 7.0 (GraphPad Software, Inc., San Diego, CA, USA). Differences between treatment means were evaluated by one-way ANOVA followed by the Duncan multiple comparison test. A probability ≤ 0.05 was considered statistically significant.

## 3. Results

### 3.1. Proline Enhances Cell Viability in pTr2 Cells

Compared with the control group (0 mmol/L proline addition), 0.50 and 1.0 mmol/L proline addition stimulated cell viability (*p* < 0.05) by 19.7% and 9.6%, respectively, at 24 h posttreatment (Figure 1). Consistently, this effect of proline was further observed (*p* < 0.05) at 48 h posttreatment, because 0.50 and 1.0 mmol/L proline addition enhanced (*p* < 0.05) cell viability by 32.7% and 26.0%, respectively, when compared with the control (Figure 1). Cell viability did not differ (*p* > 0.05) between 0 and 0.25 mmol/L proline addition at 24 h posttreatment (Figure 1). Conversely, 0.25 mmol/L proline addition stimulated cell growth (*p* < 0.05) by 19.4% when compared with 0 mmol/L proline addition at 48 h posttreatment (Figure 1).

### 3.2. Proline Activates mTORC1 Cell Signaling Pathway in pTr2 Cells

Western blotting was performed to determine the activation of mTORC1 signaling in pTr2 cells. As shown in Figure 2, protein abundance of total mTORC1 (Figure 2A), total ribosomal p70S6K1 (Figure 2B), and S6 (Figure 2C) did not differ (*p* > 0.05) among groups. Conversely, the abundance of total 4E-BP1 was reduced (*p* < 0.05) by proline addition (Figure 2D). Addition of 0.5 mmol/L proline to the culture medium enhanced (*p* < 0.05) the abundance of phosphorylated (p)-mTORC1 (Figure 2A), p-p70S6K1 (Figure 2B), p-S6 (Figure 2C), and p-4E-BP1 (Figure 2D) compared with the control (0 mmol/L proline addition), suggesting the stimulation of cell proliferation by proline. Additionally, addition of 0.25 mmol/L proline to the medium enhanced (*p* < 0.05) the abundance of p-mTORC1 compared with the control, but not by a higher concentration of proline (1.0 mmol/L). Furthermore, 0.25 mmol/L and 1.0 mmol/L proline addition enhanced (*p* < 0.05) the abundance of p-p70S6K and p-S6. The abundance of p-4E-BP1 was not affected (*p* > 0.05) by 0.25 mmol/L and 1.0 mmol/L of proline.

### 3.3. Proline Enhances Proline Transporter Expression and Catabolism in pTr2 Cells

Amino acid transporters are membrane-bound transport proteins that mediate active transport of amino acids into and out of cells [43,44]. Thus, mRNA expression of solute carrier family (*SLC*) *6A20*, *SLC36A1*, *SLC36A2*, *SLC38A1*, and *SLC38A2* in pTr2 cells was determined by real-time PCR. Proline addition dose-dependently enhanced (*p* < 0.05) the mRNA expression of *SLC6A20*, *SLC36A1*, *SLC36A2*, *SLC38A1*, and *SLC38A2* (Figure 3A). Furthermore, proline concentration in pTr2 cells was detected by HPLC. The results showed that proline treatment dose-dependently enhanced (*p* < 0.05) proline concentration in pTr2 cells (Figure 3B). PRODH, also known as proline oxidase, is an enzyme which oxidizes proline into pyrroline-5-carboxylate (P5C). Next, we investigated the protein abundance of PRODH and p-AMPK*α*. Western blot analysis demonstrated that proline added to the cell culture enhanced (*p* < 0.05) protein abundance of PRODH (Figure 3C). In contrast, 0.25 or 0.5 mmol/L proline in medium resulted in lower (*p* < 0.05) abundance of p-AMPK*α* when compared with the control (Figure 3D). However, a difference in p-AMPK*α* abundance was not observed when exogenous proline addition reached 1.0 mmol/L compared with the control (Figure 3D).

### 3.4. Proline Reduces ROS Production in pTr2 Cells

Flow cytometry analysis showed that proline addition led to (*p* < 0.05) a lower ROS level when compared with the control group (0 mmol/L) in pTr2 cells (Figure 4A). Consistent with the ROS level, mRNA expression of the catalytic subunit of glutamate–cysteine ligase (*GCLC*) and glutathione synthetase (*GSS*) was enhanced by proline addition when compared with the control (Figure 4B). Furthermore, 0.5 mmol/L proline addition elevated (*p* < 0.05) T-GSH and GSH concentration in pTr2 cells compared with the control (Figure 4C). Moreover, 0.25 mmol/L and 1.0 mmol/L proline addition increased T-GSH and GSH concentration in pTr2 cells but not significantly when compared with the control (Figure 4C).

## 4. Discussion

pTr2 cells are a cell line model widely used in reproductive biology research [9,11,12]. In the present study, we found that proline activated mTORC1 signaling pathway and reduced intracellular ROS level through enhancing GSH production. To our knowledge, this is the first study showing biochemical, cellular, and molecular evidence of proline in activating cell growth and modulating the intracellular redox environment.

The trophectoderm are the first epithelial cells to form during preimplantation development, and these cells lead to the formation of placenta and fetal membranes [13]. The placenta is essential for providing nutrients (including water) and oxygen from mother to fetus, which is vital for fetal survival, growth, and development [45]. Additionally, previous studies reported that placenta insufficiency is associated with fetal death and IUGR [25]. Thus, the growth and development of the trophectoderm are determinants for fetal survival, growth, and development [46]. Indeed, providing dietary supplementation of arginine, glutamine, and proline improves placental growth, fetal survival, and growth [45]. Restricting amino acid availability during gestation has been documented to impair fetal survival and development due to downregulation of placental mTORC1 signaling [47,48]. Additionally, emerging evidence indicates that addition of arginine, glutamine, leucine, and putrescine to culture medium enhances the growth of pTr2 cells via activating the mTORC1 signaling pathway [11]. Proline, an imino acid with various biological functions and a member of the arginine family of amino acids, is the major amino acid for polyamine synthesis in the porcine placenta because of the lack of arginase [46]. In the present study, we found that proline at 0.25 and 0.50 mmol/L enhanced pTr2 cell growth by 32.7% and 26.0% respectively, compared with the control at 48 h posttreatment (Figure 1). However, when exogenous proline addition reached 1.0 mmol/L, no further increase in cell viability was observed (Figure 1). Thus, an optimal addition of proline appears to be 0.50 mmol/L based on the cell viability. Amino acids and amino acid receptors have been shown to activate the mTOR signaling pathway. Activation of mTOR by nutrients, growth factors, energy status, and stress regulates cellular processes including growth, differentiation, and metabolism [49,50]. Further research revealed that proline enhanced the protein abundance of p-mTORC1 (Figure 2A), p-p70S6K (Figure 2B), p-S6 (Figure 2C), and p-4EB-BP1 (Figure 2D) at 0.5 mmol/L. These results are consistent with the previous studies which reported that proline addition in pluripotent cell culture activates mTORC1 signaling involved in growth and differentiation [26,51]. Additionally, in vivo data demonstrated that dietary proline supplementation enhanced protein abundance of p-mTORC1, p-p70S6K, and p-4EB-BP1 in the placenta [30]. Furthermore, increased protein synthesis was observed in the muscle, skin, and small intestine of neonatal Yucatan Miniature piglets when the diet was supplemented with proline [52]. These data indicate a role of proline in stimulating cell growth and mTOR activation.

Membrane transport of proline has received considerable attention in basic and pharmaceutical research recently. The transport of proline into cells is mediated by a number of proline transporters [44,53]. The placenta grows very rapidly during early gestation in gilts (placental mass increases by 826-fold between days 20 and 60 of gestation). Additionally, concentration of proline increases from 335 nmol/g tissue to 477 nmol/g tissue from day 20 to 60 of gestation [54], which indicates increasing proline availability is a requirement for the optimal growth of the placenta. In agreement with a previous study [51], we observed enhanced mRNA expression for proline transporters, *SLC6A20*, *SLC36A1*, *SLC36A2*, *SLC38A1*, and *SLC38A2* (Figure 3A), compared with the control. Consistent with the enhanced mRNA expression for proline transporters, HPLC analysis showed that proline addition resulted in dose-dependently increased proline concentration in pTr2 cells (Figure 3B). Proline serves as a nitrogenous substrate for endogenous synthesis of arginine, glutamate, and polyamines via P5C through oxidizing proline by PRODH, also known as proline oxidase [28,29,55]. Therefore, PRODH is a key enzyme in the process of proline metabolism [26]. In our study, proline addition dose-dependently enhanced the protein abundance of PRODH (Figure 3C). Additionally, emerging evidence reported that PRODH is essential in proline protection against hydrogen peroxide-mediated cell death [26]. Catabolism of proline in the mitochondrial compartment by PRODH will generate ATP and ROS which may act as signaling molecules [56]. Previous studies demonstrated that proline reduces the ROS level by reacting with singlet oxygen (^1^O_2_), H_2_O_2_, and OH• (pH 7–8) to form stable free radical adducts of proline [32,33,34]. In the present study, we found that 0.25 or 0.5 mmol/L proline addition led to a reduced phosphorylation level of AMPK*α*, and 1.0 mmol/L proline addition resulted in activation of p-AMPK*α*. These results may corroborate previous studies demonstrating that low levels of ROS stimulated mTOR activation, whereas accumulation of ROS through the oxidation of proline reduced the protein abundance of p-mTOR, p-S6, and p-4EBP1 expression through activation of AMPK [57,58].

Early embryo development occurs in a relatively low oxygen environment which is very sensitive to reactive oxygen species because of its low antioxidant capacity [13,21,59]. As placentation progresses, oxygen transfer increases dramatically which leads to elevated production of ROS [59]. Of note, a loss of the balance between ROS generation and antioxidant scavenging capacity has been associated with embryo/fetus death and placenta insufficiency [1,12]. Recently, new roles for proline in altering the redox status of mammalian cells have also been discovered [27,31,32]. In the present study, we found that proline addition to the cell culture reduced the intracellular ROS level, confirmed by flow cytometry analysis (Figure 4A). Previous research demonstrated that proline content inversely correlates with ROS levels [27]. Interestingly, proline addition increased GSH concentration in pTr2 cells which coordinately neutralized the free radicals and therefore reduced intracellular ROS level. These results were in agreement with previous studies which reported that proline led to increased T-GSH and protection of reduced intracellular GSH through reacting with singlet oxygen (^1^O_2_), H_2_O_2_, and OH• (pH 7–8) to form stable free radical adducts of proline [32,33,34]. Proline catabolism was proposed to generate transient ROS signals that activate nuclear factor erythroid 2-related factor 2 (Nrf2), leading to increased expression of antioxidant enzymes and life span [56]. In addition, the enhanced availability of GSH was associated with transcriptional upregulation of the catalytic subunit of *GCLC* and *GSS*, two genes involved in GSH biosynthesis (Figure 4B). These data suggest that proline accumulation appears to protect the glutathione redox state of the cell, perhaps by directly scavenging ROS.

## 5. Conclusions

The present study conducted with pTr2 cells demonstrated that proline addition stimulates the growth of placental cells and modulates the intracellular redox environment. These novel findings support a potential role of proline in the improvement of fetal–placental growth and development in pigs. Future studies that aim to elucidate the underlying mechanisms by which proline catabolism influences the mTORC1 signaling pathway should reveal new insights into mitochondrial-dependent signaling.

## Figures and Tables

**Figure 1 biomolecules-11-00742-f001:**
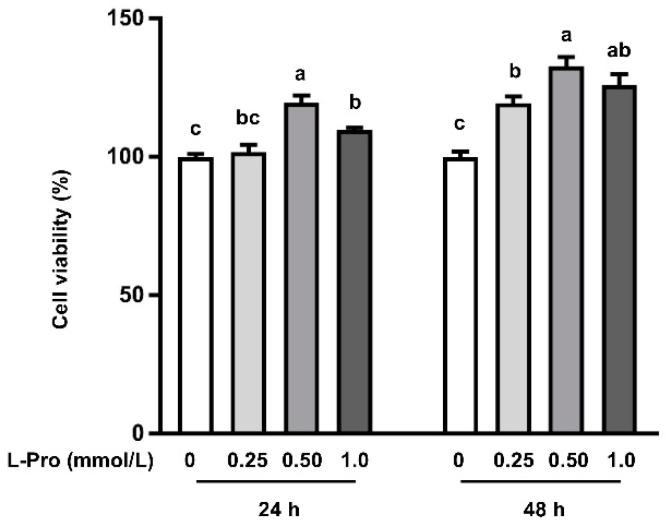
Effects of L-proline on the growth of pTr2 cells at 24 or 48 h. Data are expressed as the percentage of control group. Values are means ± SEMs, *n* = 6. Means at a time point without a common letter differ, *p* < 0.05. pTr2, porcine trophectoderm cell line 2.

**Figure 2 biomolecules-11-00742-f002:**
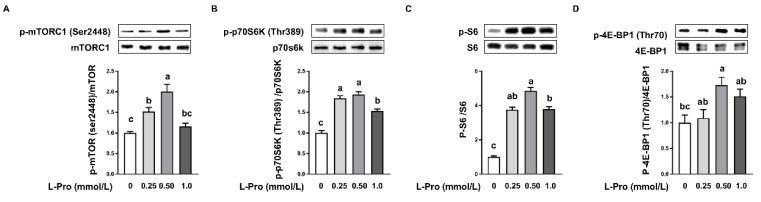
Effects of proline on mTORC1 signaling pathway in pTr2 cells. (**A**) The protein abundance of total and p-mTORC1 in pTr2 cells. (**B**) The protein abundance of total and p-p70S6K in pTr2 cells. (**C**) The protein abundance of total and p-S6 in pTr2 cells. (**D**) The protein abundance of total and p-4E-BP1 in pTr2 cells. Data are expressed as the percentage of control group. Values are means ± SEMs, *n* = 3. Means at a time point without a common letter differ, *p* < 0.05. 4E-BP1, eukaryotic translation initiation factor 4E-binding protein 1; mTORC1, mammalian target of rapamycin complex 1; p, phosphorylated; p70S6K, ribosomal protein S6 kinase; pTr2, porcine trophectoderm cell line 2.

**Figure 3 biomolecules-11-00742-f003:**
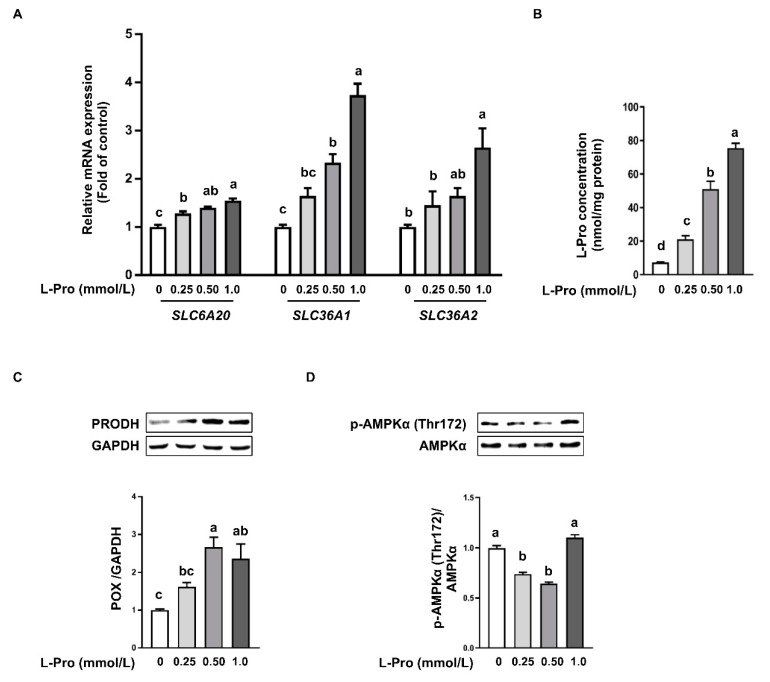
Effect of proline on proline transport and metabolism in pTr2 cells. (**A**) mRNA expression for proline transporters. (**B**) Proline concentration in pTr2 cells. (**C**) The protein abundance of PRODH in pTr2 cells. (**D**) The protein abundance of total and p-AMPK in pTr2 cells. Data are expressed as the percentage of control group. Values are means ± SEMs, *n* = 3. Means at a time point without a common letter differ, *p* < 0.05. GAPDH, glyceraldehyde-3-phosphate dehydrogenase; PRODH, proline dehydrogenase; pTr2, porcine trophectoderm cells 2; SLC, solute carrier family.

**Figure 4 biomolecules-11-00742-f004:**
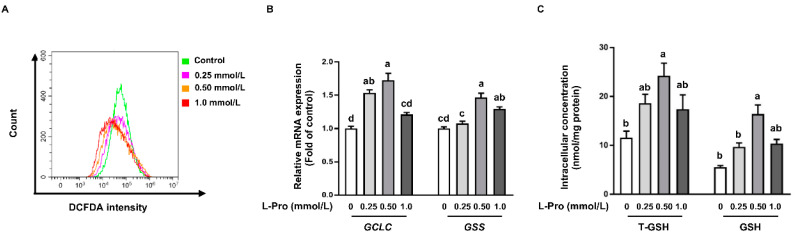
Effect of proline on ROS level and GSH concentration in pTr2 cells. (**A**) DCFH-DA positive populations were determined by flow cytometry analysis in a time course. (**B**) mRNA levels for *GCLC* and *GSS*. (**C**) Intracellular T-GSH and GSH. Data are expressed as the percentage of control group. Values are means ± SEMs, *n* = 3. Means at a time point without a common letter differ, *p* < 0.05. DCFH-DA, 2′,7′-Dichlorofluorescein diacetate; pTr2, porcine trophectoderm cells 2; GCLC, catalytic subunit of glutamate–cysteine ligase; GCLM, glutamate–cysteine ligase modifier subunit; GSH, glutathione; GSS, glutathione synthetase.

**Table 1 biomolecules-11-00742-t001:** Primers used for real-time PCR.

Genes	Accession No.	Primers
*GAPDH*	NM_001206359.1	F: 5′-GTCGGAGTGAACGGATTTGG-3′
		R: 5′-AGTGGAGGTCAATGAAGGGG-3′
*GCLC*	XM_003482164.4	F: 5′-GAAAACCAGGCTCTCTGCAC-3′
		R: 5′-ATCGCTTCGTCTGGAAAGAA-3′
*GSS*	NM_001244625.1	F: 5′-GGCTGAAGGACAGTGAGGAG-3′
		R: 5′-TTCCCTGCCTGACATAGACC-3′
*SLC6A20*	XM_021068640.1	F: 5′-TGGTGGTGTCCTTCTTCCTC-3′
		R: 5′-ATTCAGTGGGCAGACAGACC-3′
*SLC36A1*	XM_021077073.1	F: 5′-CATCGGCATCTTCTTCACCT-3′
		R: 5′-GGTCTATCACCAGCCTCCAA-3′
*SLC36A2*	XM_021077082.1	F: 5′-TTGCTAGCCATGGGCTTCAT-3′
		R: 5′-AGAAGCTCACGATACGCCTTC-3′
*SLC38A1*	XM_003355629.4	F: 5′-GAACACTGGAGCAATGCTGA-3′
		R: 5′-ATAGCCGAGATAGCCCAGGT-3′
*SLC38A2*	XM_005659640.3	F: 5′-CCTACCTCCTGACAGCTTGC-3′
		R: 5′-AACCAAAAGCACCAAACAGG-3′

## Data Availability

Data are contained within the article.

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
