# Peer review of "L-Proline Activates Mammalian Target of Rapamycin Complex 1 and Modulates Redox Environment in Porcine Trophectoderm Cells"

_biomolecules, 2021, doi:10.3390/biom11050742_

Round 1

Reviewer 1 Report

This is a useful study that explores the relation between proline supplementation and cellular physiology in pTr2 cells. The study is mostly descriptive because it lacks manipulation of the system (other than proline supplementation) to support its arguments. As a result, the conclusions are based on correlation, which is not a strong as causation.

Some suggestions to improve the study are made in the following:

156 and Fig.1: The viability assay is based on the presence of electron donors to reduce the day. Given that metabolism of proline generates electron donors, the increase in viability might just be an increase of reducing equivalents. The authors equate cell growth with cell viability, which is not the same.

205 The listed transporters may not be the most relevant for proline transport in this cell line. Most cells accumulate proline via SNAT1 and SNAT2. It would be important to provide Western blot data to substantiate the roles of SLC6A20 and SLC36A2. SLC36A1 is a lysosomal transporter in most cells and surface-biotinylation would be required to suggest a role in proline uptake.

273 Text was cut off by the image.

283/284 The cell line is not similar to human beings, it may be similar to human trophoectoderm cells.

 329-330 "Consistently with the enhanced mRNA expression for proline transporters, HPLC analysis showed that proline addition resulted in dose.
dependently increased proline concentration in pTr2 cells". This statement is not justified. The increase of intracellular proline is most likely driven by the elevated extracellular concentration. To substantiate this argument the cell line with induced transporter expression would need to be placed in the same medium as a control cell line in which transporters are expressed at base level. A comparison of cellular amino acid levels would then reveal whether there is enhanced accumulation.

338 and 355: The discussion is confusing. The authors make the case for enhanced ROS production by proline metabolism and for ROS scavenging by proline. It is not convincing to use the argument in both directions to support the data. Please improve the discussion around ROS.

Author Response

Dear Reviewer:

Thanks for your comments on our manuscript (Biomolecules-1191371) entitled “L-Proline activates mammalian target of rapamycin complex 1 and modulates redox environment in porcine trophectoderm cells”.

We have carefully considered all the comments of Reviewers on our manuscript and

provided our response as shown below. We added data on mRNA expression of SNAT1 and SNAT2. We did not detect protein abundance of SLC6A20, SLC36A1, and SLC36A2 and those antibodies are currently not available in our lab. The materials and methods section, as well as results and discussion section were revised accordingly. All the revisions have been highlighted in this version.

Our point-by-point responses are summarized as follows:

Reviewer #1:

Comments and Suggestions for Authors

This is a useful study that explores the relation between proline supplementation and cellular physiology in pTr2 cells. The study is mostly descriptive because it lacks manipulation of the system (other than proline supplementation) to support its arguments. As a result, the conclusions are based on correlation, which is not a strong as causation.

Reply:

Thanks for reviewing our manuscript and the constructive comments. Your comments and suggestions are of great value for our study.

Some suggestions to improve the study are made in the following:

  1. 156 and Fig.1: The viability assay is based on the presence of electron donors to reduce the day. Given that metabolism of proline generates electron donors, the increase in viability might just be an increase of reducing equivalents. The authors equate cell growth with cell viability, which is not the same.

Reply:

Thank you for your suggestions. The viability assay was done by using CCK-8 according to published papers (Toxins (Basel). 2019 Apr 16;11(4):226ï¼›J Psychopharmacol. 2010 Jul;24(7):1055-67). Cell Counting Kit-8 (CCK-8) is a convenient and sensitive method for the research of cell number determination and cell proliferationtotoxicity assay than previous ways. The kit utilizes a highly water-soluble tetrazolium salt, WST-8, which produces a water-soluble formazan dye upon reduction in the presence of an electron mediator. The amount of the formazan generated by dehydrogenases is directly in proportion to the numbers of living cells. The detection sensitivity by CCK-8 is higher than other tetrazolium salts such as MTT, XTT, MTS or WST-1.

  1. 205 The listed transporters may not be the most relevant for proline transport in this cell line. Most cells accumulate proline via SNAT1 and SNAT2. It would be important to provide Western blot data to substantiate the roles of SLC6A20 and SLC36A2. SLC36A1 is a lysosomal transporter in most cells and surface-biotinylation would be required to suggest a role in proline uptake.

Reply:

Thanks for your constructive suggestion. The proton-coupled amino acid transporters (PAT) constitute the recently identified SLC36 family of mammalian membrane transporters. PAT1 (SLC36A1) and PAT2 (SLC36A2) were identified from mouse intestine and embryonic tissue. Previous study demonstrated that PAT1 is the major transport system that mediates the concentrative (uphill) uptake of L-proline into Caco-2 cells (J Physiol. 2003 Jan 15;546(Pt 2):349-61). Free proline is absorbed from the lumen by the amino acid symporter PAT1 (SLC36A1) or the imino acid symporter SIT1 (SLC6A20). The mRNA of mouse PAT1 is highly expressed in small intestine, colon, kidney and brain but also in lung, liver and spleen. The PAT2 is mainly found in heart and lung. Significant expression was also observed in kidney, testes, liver and spleen. L-Proline is also a substrate for system SNAT2 (amino acid transporter A2, SLC38A2), a subtype of amino acid transport system A. Competition experiments using radioactive labeled MeAIB revealed that SNAT2 is able to transport several neutral amino acids such as alanine, glycine, serine, proline, methionine, asparagine, glutamine, threonine and leucine. System A is involved in efflux transport of L-proline across the blood brain barrier. From a biopharmaceutical point of view, proline transporters are very relevant members of the solute carrier families. In addition to specific transport, simple diffusion of proline across epithelial barriers might also play a significant role.

Antibodies for SLC6A20 and SLC36A2 are currently not available in our lab. Furthermore, we measured the expression of SNAT1 (SLC38A1) and SNAT2 (SLC38A2) in the revised version, please see details in line 210-214 and Figure 3.

  1. 273 Text was cut off by the image.

Reply:

Sorry for that. We have revised this image, please see details in line 264-273.

  1. 283/284 The cell line is not similar to human beings, it may be similar to human trophoectoderm cells.

Reply:

Sorry for the confusion. We revised it in this version, please see line 291-295 for details.

  1. 329-330 "Consistently with the enhanced mRNA expression for proline transporters, HPLC analysis showed that proline addition resulted in dose. dependently increased proline concentration in pTr2 cells". This statement is not justified. The increase of intracellular proline is most likely driven by the elevated extracellular concentration. To substantiate this argument the cell line with induced transporter expression would need to be placed in the same medium as a control cell line in which transporters are expressed at base level. A comparison of cellular amino acid levels would then reveal whether there is enhanced accumulation.

Reply:

Thanks for your suggestion. Apical ion gradient-coupled transporters such as PAT1 and SIT1 accumulate their substrates in the cell, i.e. they transport uphill. Compared to their contribution, proline uptake by simple diffusion might be negligible.

  1. 338 and 355: The discussion is confusing. The authors make the case for enhanced ROS production by proline metabolism and for ROS scavenging by proline. It is not convincing to use the argument in both directions to support the data. Please improve the discussion around ROS.

Reply:

Sorry for the confusion. We improved the discussion around ROS, please see details in line 351-359.

Thank you.

Zhenlong Wu

Reviewer 2 Report

To authors,

I believe that the paper is well written and the experiments were performed adequately. My concern is the followings.

Conclusion of my concern: Whether this study accounts for “pig” placenta (materno-placental-fetal function/relationship) OR it can be applied to that of “human” is obscure. If the former is the case, less attentions may be paid by the readers compared with the latter case. i) you used porcine blastocyst but your targets (future/real) are human placenta, then state this meaning straightforwardly. ii) if this is the study only for pig placenta and whether or not this data can be applied to (also represent) humans is unclear, then, state this meaning. Please take either strategy.  

  1. You stated , “We have recently reported that maternal proline supplementation during gestation enhanced fetal survival, reproductive performance, placental growth, and amino acid 282 transport from dam to fetus [35]”. This meaning (or sentence itself) should be placed (moved to) Introduction. Please strengthen the importance (worthy to study) of proline.
  2. You state, “pTr2 cells, a cell line model widely used in reproductive biology research due to its similarities with human beings”: depending on the strategy of i) or ii), please strengthen this meaning in Introduction section.
  3. You state, “These novel findings support a potential role of proline to improve fetal-placental growth and development in mammals, including humans and pigs”: Please be consistent to (agree with) the strategy i) or ii).
  4. As you know, the human placenta is hemo-(tri) chorial placenta. Human trophoblasts can be divided into syncytiotrophoblast and cytotrophoblast, with the latter further being classified into two (extrachorial (extravillous)) or ordinary (intravillous). All these have completely different characters and biological meaning. If you consider that the present study, more or less, reflect that of “human”, then, please definitely state “what type of human trophoblasts you assumed/considered”. “human placenta” or “human trophoblast” is too vague. One cannot conclude (without explanation) anything on “human” placenta based on the present data “without reasonable explanations”. In doing so, please do not much expand the volume. You need not write much to explain things.

Author Response

Dear Reviewer:

Thanks for your comments on our manuscript (Biomolecules-1191371) entitled “L-Proline activates mammalian target of rapamycin complex 1 and modulates redox environment in porcine trophectoderm cells”.

We have carefully considered all the comments of Reviewers on our manuscript and

provided our response as shown below. Data in our paper can be applied to human is unclear, so we paid attention on pig placenta rather than human placenta, however the underlying mechanism is similar among different trophoblast in mammals. These novel findings support a potential role of proline to improve fetal-placental growth and development in pigs. The materials and methods section, as well as results and discussion section were revised accordingly. All the revisions have been highlighted in this version.

Our point-by-point responses are summarized as follows:

I believe that the paper is well written and the experiments were performed adequately. My concern is the followings.

Reply:

Thanks for supportive comments on our study.

Conclusion of my concern:

  1. Whether this study accounts for “pig” placenta (materno-placental-fetal function/relationship) or it can be applied to that of “human” is obscure. If the former is the case, less attentions may be paid by the readers compared with the latter case. i) you used porcine blastocyst but your targets (future/real) are human placenta, then state this meaning straightforwardly. ii) if this is the study only for pig placenta and whether or not this data can be applied to (also represent) humans is unclear, then, state this meaning. Please take either strategy.

Reply:

Thanks for reviewing our manuscript and the constructive comments. Your comments and suggestions are of great value for our study. Data in our paper can be applied to human is unclear, so we paid attention on pig placenta rather than human placenta, please see details in the revised version.

  1. You stated, “We have recently reported that maternal proline supplementation during gestation enhanced fetal survival, reproductive performance, placental growth, and amino acid transport from dam to fetus [35]”. This meaning (or sentence itself) should be placed (moved to) Introduction. Please strengthen the importance (worthy to study) of proline.

Reply:

The suggested change was made, thanks!

  1. You state, “pTr2 cells, a cell line model widely used in reproductive biology research due to its similarities with human beings”: depending on the strategy of i) or ii), please strengthen this meaning in Introduction section.

Reply:

The suggested change was made, please see details in line 46-47.

  1. You state, “These novel findings support a potential role of proline to improve fetal-placental growth and development in mammals, including humans and pigs”: Please be consistent to (agree with) the strategy i) or ii).

Reply:

Sorry for the confusion. We re-wrote the sentences for clarity. Please see Line 378 and 379 for detail.

  1. As you know, the human placenta is hemo-(tri) chorial placenta. Human trophoblasts can be divided into syncytiotrophoblast and cytotrophoblast, with the latter further being classified into two (extrachorial (extravillous)) or ordinary (intravillous). All these have completely different characters and biological meaning. If you consider that the present study, more or less, reflect that of “human”, then, please definitely state “what type of human trophoblasts you assumed/considered”. “human placenta” or “human trophoblast” is too vague. One cannot conclude (without explanation) anything on “human” placenta based on the present data “without reasonable explanations”. In doing so, please do not much expand the volume. You need not write much to explain things.

Reply:

Thanks for your suggestion. The swine model has physiological and physical similarities to human beings. Scientific evidence supports this similarity that most systems and organs have between pigs and human beings. They are good models for specific pathologies and there is a good knowledge of the animal. Porcine trophectoderm cells were isolated using nonenzymatic dispersion of trophoblast from conceptuses collected on day 12 of gestation. Trophectoderm cells are critical to implantation because they must communicate with and adhere to maternal uterine epithelium. pTr2 line as an in vitro model for studying the functional biology of pig trophectoderm, which is an early and critical participant in establishing blastocyst interactions with the uterine luminal epithelium. Intermediate filament cytokeratin-7 (KRT7) is highly expressed throughout in porcine trophoblast. It has been used to assess the purity of human placental villous trophoblast cells by flow cytometry. Actually, few studies report the similarity pig and human trophoblast (Endocrinology. 2001 Jun;142(6):2303-10; Endocrinology. 2005 Sep;146(9):3933-42.), however the underlying mechanism is similar among different trophoblast in mammals.

Thank you.

Zhenlong Wu

Round 2

Reviewer 1 Report

The revisions to the manuscript are acceptable.